# LRM-Zero: Training Large Reconstruction Models with Synthesized Data

**Desai Xie**[† 1, 2]     **Sai Bi**[1]     **Zhixin Shu**[1]     **Kai Zhang**[1]     **Zexiang Xu**[1]
**Yi Zhou**[1]     **Sören Pirk**[3]     **Arie Kaufman**[2]     **Xin Sun**[1]     **Hao Tan**[1]

[1]Adobe Research     [2]Stony Brook University     [3]Kiel University
`dexxie,ari@cs.stonybrook.edu`
`sbi,zshu,kaiz,zexu,yizho,xinsun,hatan@adobe.com`
`soeren.pirk@gmail.com`

## Abstract

We present *LRM-Zero*, a Large Reconstruction Model (LRM) trained entirely on synthesized 3D data, achieving high-quality sparse-view 3D reconstruction. The core of *LRM-Zero* is our procedural 3D dataset, *Zeroverse*, which is automatically synthesized from simple primitive shapes with random texturing and augmentations (e.g., height fields, boolean differences, and wireframes). Unlike previous 3D datasets (e.g., Objaverse) which are often captured or crafted by humans to approximate real 3D data, *Zeroverse* completely ignores realistic global semantics but is rich in complex geometric and texture details that are locally similar to or even more intricate than real objects. We demonstrate that our *LRM-Zero*, trained with our fully synthesized *Zeroverse*, can achieve high visual quality in the reconstruction of real-world objects, competitive with models trained on Objaverse. We also analyze several critical design choices of *Zeroverse* that contribute to *LRM-Zero*'s capability and training stability. Our work demonstrates that 3D reconstruction, one of the core tasks in 3D vision, can potentially be addressed without the semantics of real-world objects. The *Zeroverse*'s procedural synthesis code and interactive visualization are available at: https://desaixie.github.io/lrm-zero/.

## 1 Introduction

The current wave of rapid development in foundation models [10] has been empowered by two key components: scalable model architectures [86, 40, 67] and massive datasets [19, 68, 79]. Foundation models in text [65, 3, 85], image [70, 29] and video [48, 5, 35] domains have capitalized on both factors. As there is a unified trend of adopting transformers [86] as the model architecture, many recent works have identified the training data as the most crucial factor [8, 20, 37, 2, 11].

Following the progress in other modalities, transformer-based [86] 3D Large Reconstruction Models [41, 49, 90, 98, 84] (LRM) has emerged as a potential foundation model for the 3D domain, which uses a scalable model architecture. However, 3D data is still difficult to acquire, considering that both manually capturing and hand-crafting 3D objects are expensive, time-consuming and require special expertise. Also, 3D data is more sensitive for its license safety, different types of bias, and identity leakage.

---

†This work is done while Desai is an intern at Adobe Research.

38th Conference on Neural Information Processing Systems (NeurIPS 2024).

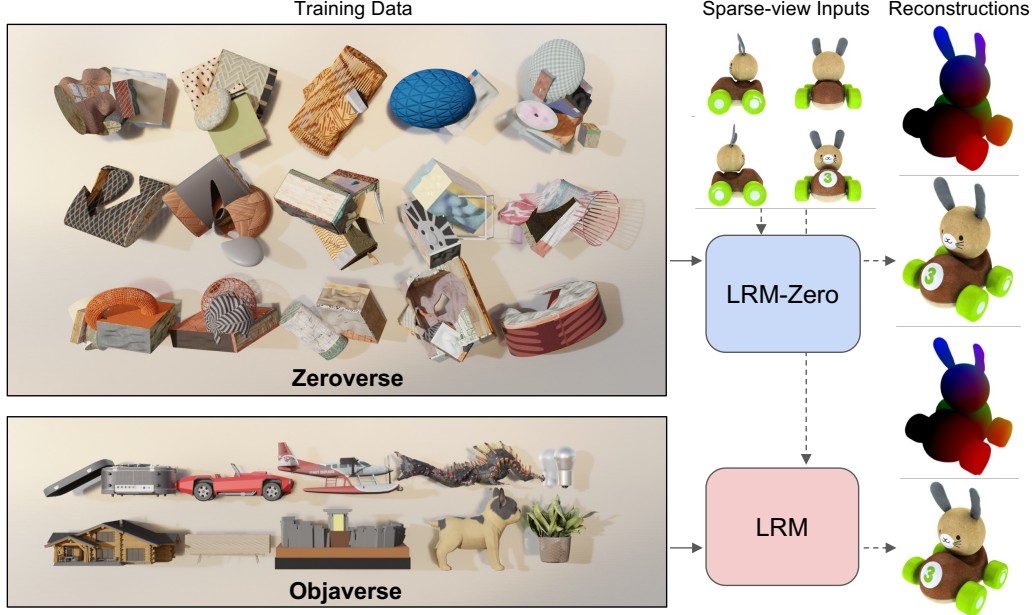

Training Data    Sparse-view Inputs    Reconstructions

Zeroverse

Objaverse

LRM-Zero

LRM

Figure 1: We present our *LRM-Zero* framework trained with synthesized procedural data *Zeroverse*. *Zeroverse* (top left) is created from random primitives with textures and augmentations, thus it does not contain semantical information as in Objaverse (bottom left). Nevertheless, when training with the same large reconstruction model architecture [107] on both datasets, *LRM-Zero* can match objaverse-trained LRM's (denoted as 'LRM') visual quality (right part) of reconstructions. A possible explanation is that 3D reconstruction, although serves as a core task in 3D vision, rely mostly on local information instead of global semantics. Reconstruction is visualized with RGB and position-based renderings, and interactive viewers can be found on our website.

Thus, in this paper, we propose *LRM-Zero*, trained on purely synthesized data, to explore another route which can potentially resolve the 3D data scarcity, licensing, and bias issues. The name 'Zero' highlights our synthesized and non-semantic training data, which we named as *Zeroverse*. *Zeroverse* is a procedural, amorphic alternative to Objaverse [23] in training reconstruction models. The comparison between *Zeroverse* and Objaverse is illustrated in Fig. 1, and more visual comparisons can be found in Appendix. The data in *Zeroverse* is procedurally created by randomly composing primitive shapes with textures and applying shape augmentations. The process resembles the previous work Xu et al. [100]. We select five primitive shapes: cube, sphere, cylinder, cone, and torus to cover different types of surfaces and topological characteristics. The textures are randomly applied, which is realistic at low-level but do not contain high-level semantics. The three different augmentation methods, i.e., height-field, boolean difference, and, wireframes, help increase the data diversity and add more curvatures, concavity, and thin structures, respectively. The primitive shapes, textures, and an illustration of the augmentations are shown in Fig. 2. In this work, we experiment with 400K *Zeroverse* data, which roughly matches the number of meaningful data in Objaverse (i.e., excluding rendering failures, flatten 3D data, point clouds, unsafe data from the overall 800K data). The initial experiments indicate that further increasing the amount of data is not effective, and we refer the reader to Appendix for our early results on scaling the data size.

We validate our *Zeroverse* by training GS-LRM [107] over it, and we denote this model as *LRM-Zero*. Surprisingly, we found that *LRM-Zero* can achieve a reconstruction quality similar to that of GS-LRM trained on Objaverse, seeing Fig. 1. More comparisons are provided in the Appendix. We also quantitatively evaluate the model on two standard 3D reconstruction benchmark ABO [18] and GSO [28]. For sparse-view reconstruction (i.e., 4 views and 8 views), *LRM-Zero* reaches competitive results against GS-LRM, and the best results gap is as low as 1.12 PSNR, 0.09 SSIM, and 0.006 LPIPS. A plausible reason for such "zero"-shot data generalization is that 3D reconstruction (with poses) relies more on the local visual clues instead of the global semantics. This is more obvious for dense-view reconstruction (e.g. 100 input views) where single-shape optimization without any data prior can reach good results [6, 46]. For the sparse-view reconstruction that we focused, *LRM-Zero*

can possibly rely on the local details (such as cross-view patch correspondence) to infer the shape, where *Zeroverse* supports *LRM-Zero* to learn such knowledge.

We analyze the effectiveness of *Zeroverse*'s design, especially for shape augmentations. We find that each type of augmentation provides visible structural improvements for the reconstructions, and most of the improvements are reflected in the metrics of our benchmarks. We also study the impact of different dataset designs on another critical property of *LRM-Zero*: training stability. Training stability is crucial for large-scale training as large models are more prone to diverge after training for a significantly long time [72, 17, 22]. We empirically found that careless design of *Zeroverse* can introduce significant instability during the training of *LRM-Zero*. As both data complexity and model hyperparameters can affect the training stability, a model-data co-design is helpful in our experiments, i.e., the model's hyperparameters and data properties are tuned jointly. Lastly, we show the generalizability of both *Zeroverse* and *LRM-Zero*. For *Zeroverse*, we show that the dataset can also enable training a NeRF-based reconstruction model and reaches competitive results to Objaverse-trained models. For *LRM-Zero*, we demonstrate that the model can generalize across different datasets including realistic 3D data, such as OmniObject3D [95] and OpenIllumination [55]. We also show that *LRM-Zero* can be combined with off-the-shelf multi-view diffusion models to support both text-to-3D generation and image-to-3D generation.

The key contribution of this paper is to demonstrate that purely synthesized data can be utilized to learn generic 3D priors for sparse-view 3D reconstruction, a core task of 3D vision. While our work may appear straightforward, it provides a minimal, yet generalizable proof-of-concept which can inspire the community to exploit procedural 3D data for 3D tasks in the future. We also provide carefully crafted studies on the co-design of data and model, as well as their effect on training stability and generalization.

Lastly, we provide the interactive *Zeroverse* data visualization and *LRM-Zero* reconstruction results in our website https://desaixie.github.io/lrm-zero/. We recommend the readers to have a check. The *Zeroverse* data synthesis script is released at https://github.com/desaixie/zeroverse, and we hope that it can facilitate future research.

## 2 Background: feed-forward reconstruction model

Feed-forward 3D reconstruction targets to learn a model that can regress the 3D shapes from multi-view images. The sparse-view version of this task is illustrated in the right part of Fig. 1, where multiple input views are presented, and the output is a 3D representation. To solve this task, LRM [41] introduces a pure-transformer based method which allows scalable training. Original LRM uses NeRF [61, 12] as 3D representation, and a bunch of later works [13, 84, 16, 99, 107] extend it to Gaussian Splatting [46], which is another 3D representation proposed recently. This paper is mostly experimented with the GS-LRM [107] architecture given its simplicity in model design (i.e., a pure-transformer architecture) and the SotA reconstruction quality.

GS-LRM predict the 3D Gaussians from the $n$ multi-view images $I_1, \ldots, I_n$. The images are first patchified to features $f_1, \ldots, f_n$ with shared non-overlapping (i.e., stride equals to kernel size) convolutions. Then features are flattened and concatenated as the input to a self-attention transformer.

$$f_1, \ldots, f_n = \mathrm{Conv}(I_1), \ldots, \mathrm{Conv}(I_n) \tag{1}$$

$$x = [\mathrm{Flatten}(f_1); \ldots; \mathrm{Flatten}(f_n)] \tag{2}$$

$$y = \mathrm{Transformer}(x) \tag{3}$$

The output $y$ of the transformer will be directly interpreted as the Gaussian Splatting parameters, and serves as the representation of the output 3D object. These parameters can be rendered for training losses or viewed interactively. The GS-LRM model is purely trained with RGB rendering loss by minimizing the difference between ground truth image and the rendering images. For more details of the GS-LRM model [107] architecture and Gaussian Splatting representation [46], please refer to the original papers. After briefly introducing the backbone model architecture of *LRM-Zero*, we next introduce our procedural data *Zeroverse* to train such a model.

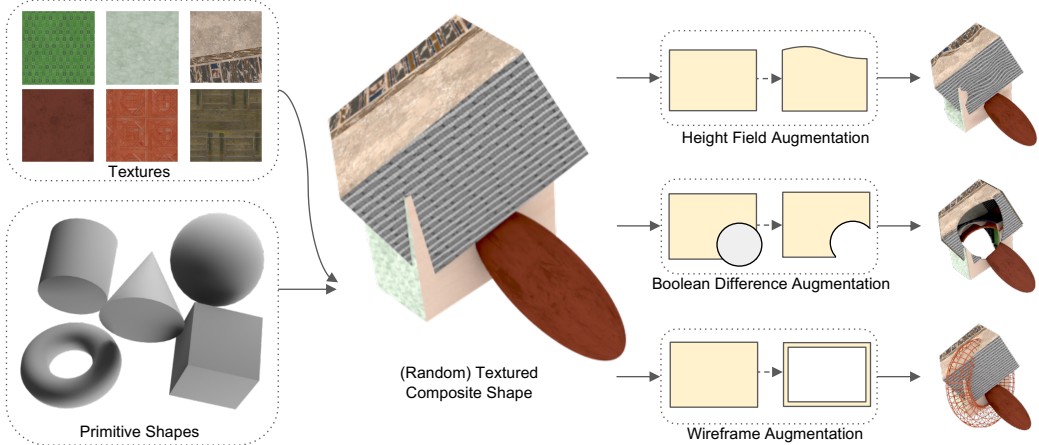

Figure 2: Illustration of the *Zeroverse* data creation process. A random textured shape is first composited from primitive shapes and textures (Sec.3.1). Then different augmentations (i.e., height field, boolean difference, wireframes in Sec . 3.2) are applied to enhance the dataset characteristics (e.g., curved surfaces, concavity, and thin structures). More visualizations in Appendix and website.

# 3 The *Zeroverse* dataset

In this section, we introduce the creation of *Zeroverse* that supports training a sparse-view large reconstruction model (LRM). *Zeroverse* consists of procedurally synthesized shapes with randomized parameters by revisiting the pipeline in the previous work [100], which was initially proposed for relighting and later extended for view synthesis [101] and material estimation [7, 52]. As illustrated in Fig. 2, the process first composites primitive shapes with random texturing (Sec. 3.1). Then, different augmentations are applied to enhance the diversity of the data (Sec. 3.2). As the LRM-based model only relies on multi-view rendering to train the model (i.e., do not require geometry supervision), the *Zeroverse* objects are always saved in the compact mesh format.

## 3.1 Composing primitives into textured shapes

**Primitive shapes.** Our synthetic object creation process starts with a pool of primitive shapes. The pool only consists of basic shapes for the 3D world. Specifically, in our implementation, we have 5 primitives: cube, sphere, cylinder, cone, and torus. Intuitively, cubes and spheres provides knowledge on the sharp straight lines and the purely curved shapes. Cylinders and cones contain different curved surfaces besides sphere. The specialty of torus is its hole, which is topologically different to the above shapes (i.e., a torus has genus 1). Although it is possible to create holes through combinations and augmentations (e.g., the boolean difference and wireframe in Sec. 3.2), we decide to explicitly add this capacity to our dataset. Also, the combination of multiple torus is easy to create shapes with higher genus (i.e., roughly the number of disjoint holes in a connected shape).

**Compositions.** With a reasonable pool of primitive shapes, we then compose them together to construct complex shapes, offering more diverse visual cues for the reconstruction task. We randomly sample 1 to 9 primitives (with replacement) from the primitive pool. The sampling probability of the numbers of primitives is configurable. Each sampled primitive will independently be scaled, translated, and rotated randomly. We simply combine these affine-transformed shapes together without special handling of the shape intersections or disconnections. Thus it is possible to have multiple disjoint shapes in one scene, which we will still refer to as one object. This satisfies the requirement for real-world reconstruction applications, where simple disjoint shapes would be considered as a single object.

**Texturing.** For each surface of the shape, we apply a texture randomly sampled from an internal dataset. To support the research community, in our public release version, we provide an alternative public texture dataset.

Table 1: Quantitative results comparing *LRM-Zero* with GS-LRM [107] (trained on Objaverse) under the 8-input-view setting. We use GSO [28] and ABO [18] evaluation datasets and PSNR, SSIM, and LPIPS [108] metrics. *LRM-Zero* demonstrates competitive performance against GS-LRM.

| | | | GSO | | | ABO | | |
|---|---|---|---|---|---|---|---|---|
| | | | PSNR ↑ | SSIM ↑ | LPIPS ↓ | PSNR ↑ | SSIM ↑ | LPIPS ↓ |
| 8 input views | Res-512 | GS-LRM | **33.23** | **0.971** | **0.031** | **30.92** | **0.944** | **0.067** |
| | | *LRM-Zero* | 31.62 | 0.960 | 0.039 | 28.71 | 0.929 | 0.078 |
| | Res-256 | GS-LRM | **31.90** | **0.966** | **0.030** | **30.66** | **0.949** | **0.055** |
| | | *LRM-Zero* | 30.78 | 0.957 | 0.036 | 28.82 | 0.934 | 0.065 |

## 3.2 Shape augmentations

We apply augmentation to the textured shapes to add diversity and complexity that resembles real-world objects and is not covered by the initial shape in Sec. 3.1. We implement three augmentation operators: height field, boolean difference, and wireframe conversion for better data coverage of curved shapes, concave shapes, and thin structured respectively. These diversities of the data will be reflected by the capacity of large reconstruction models with observable structural improvements (studied in Sec. 5.1). We illustrated the process and example results in the right part of Fig. 2. We do not apply the augmentation 'boolean difference' and 'wireframe' at the same time. This is for training stability (studied in Sec. 5.2) as we empirically found that an ultra-complex shape can lead the reconstruction model training to non-convergence.

**Height fields.** Most of the surfaces (except the torus) of our primitives have constant curvatures, and we apply height fields augmentation in Xu et al. [100] to break this constraint. An illustration of the height map can be found in Fig. 2 (top right). In detail, for each face of the primitives, we apply a height field with varying heights and curvatures to displace the surface vertices, making the surface curved and bumpy. Specifically, the magnitude of height is randomly sampled at each position in the map and we use bicubic interpolation to obtain smooth surfaces.

**Boolean difference.** Concave structures are common in real-world objects, for example, bowls, hats, spoons. However, the concavity is not well captured by the previous pipeline. To resolve this, we 'subtract' primitives from the shapes, which can be considering as a reversion of the 'additive' operators in the combination process. This is implemented by computing the boolean difference between the composite object in Sec. 3.1 and a basic primitive from our pool. In details, we use Blender's boolean modifier and solidify modifier to augment the initial shape. The inside faces of the resulting cut shape will have the same texture as the outside faces. Besides introducing concavity to the dataset, the boolean difference operation also expose the 'interior' of the shape (as shown in Fig. 2), which helps the reconstruction model to handle complex structures. The actual effect of the boolean operator is quite diverse, and we refer the reader to check the visualization in the Appendix.

**Wireframe.** Besides concavity, thin structures (especially the striped or repeated one) is another challenge in real-world reconstruction, for example, hairs, baskets, railings. To train a reconstruction model capable with thin structures, we want to explicitly add this characteristic to our *Zeroverse* dataset. And for simplicity, we use the wireframe. Wireframe is a basic augmentation from the primitive shapes, which generally converts their meshes to the skeletons. It is pre-implemented in multiple libraries, and we take the shape modifier in Blender. The results are illustrated in Fig. 2 (i.e., a wireframe of torus) and more in Appendix. The texture of the wireframe is inherited from the primitive shape but usually not distinguishable due to its thin surfaces.

## 4 Experiments

### 4.1 *LRM-Zero* experiment details

For rendering the multi-view images of *Zeroverse*, we follow [41]. For each object in *Zeroverse*, we render 32 views with randomly sampled camera rotations and random distances in the range of [2.0, 3.0]. Each image is rendered at the $512 \times 512$ resolution with uniform lighting. We use the same

network architecture and follow the hyperparameters/implementation (e.g., 80K training steps, details as GS-LRM [107]). We only decrease perceptual loss weight from 0.5 to 0.2 to improve training stability. For the results comparison, we pre-train the model with 256-resolution and fine-tuned on 512-resolution following GS-LRM. The overall training uses 64 A100 GPUs and takes 3 days. For analysis and ablation studies, we only run the 256-resolution experiments. Please refer to the original GS-LRM paper [107] for more experimental details.

Metric evaluations for results and analysis are mostly conducted on two relatively large benchmarks: Google Scanned Objects (GSO) [28] and Amazon Berkeley Objects (ABO) [18]. In our paper, we use 8 structural input-view as the standard evaluation protocol to increase view coverage. The 4 structural input-view results are provided in Appendix. In details, for 8 structural views, we render from 0 elevation with 0, 90, 180, 270 azimuth plus 40 elevation with 45, 135, 225, and 315 azimuth, while 4 structural views render from 20 elevation with 0, 90, 180, 270 azimuth. The testing views for metric calculation are randomly sampled. The generalization experiments in Sec. 5.3 use either 8 random input views for generalization test, or the fixed cameras provided by the generated models. We always assume that the camera poses are provided with input views.

## 4.2 Results

We evaluate *LRM-Zero* on the benchmarks and show the results in Tab. 1. The absolute PSNR values of GSO and ABO are over 30 and 28.7 respectively, which indicates that the reconstruction has high visual quality. Compared to GS-LRM [107] trained on Objaverse, the metric still shows a gap, but within a reasonable range of 1.1 PSNR on GSO and 1.9 PSNR on ABO. The gap is larger for higher resolution (i.e., Res-512) and it is possibly due to the training configuration of our 512-res fine-tuning is sub-optimal. Qualitatively, we do not observe significant visual difference between the reconstructed 3D models from *LRM-Zero* and GS-LRM. An example comparison is shown in Fig. 1 and some more comparisons in the Appendix. The interactive viewer of *LRM-Zero* reconstruction results can be found in our website.

After viewing the *LRM-Zero* visual results and the sparse-view reconstruction setup, we found that both 4-view and 8-view can not fully cover the object surfaces thus the model needs to hallucinate the invisible parts. This hallucination ability requires semantic understanding of the 3D objects while *Zeroverse* lacks by design. It might be the major reason of result gap between *Zeroverse*-trained and Objaverse-trained models in Tab. 1. The invisible regions can be mitigated by reconstructing from more views (either capturing or generating). However, more views involves more tokens and challenges the computation cost of the current fully-connected self-attention design in GS-LRM, thus beyond the scope of current paper and we leave it as future works.

## 5 Analysis

In this section, we analyze the properties of our *LRM-Zero* trained with the synthesized *Zeroverse*. We first conduct ablation studies in Sec. 5.1 to show the effectiveness of *Zeroverse* augmentations. Next, Sec. 5.2 explores stabilized training of *LRM-Zero* from both data and model perspective, as training stability is one of the key challenges in large-scale training [22, 17]. Last, we show the generalization of our methods by applying *LRM-Zero* over diverse data, and trained different reconstruction models on *Zeroverse*.

## 5.1 Ablation studies on different augmentations

We conduct ablation studies to verify the effectiveness of our height field, boolean difference, and wireframe augmentations. We show both quantative and qualitative comparisons.

**Boolean difference and wireframe augmentation.** As our sampling strategy does not apply boolean difference and wireframe augmentations jointly to avoid over-complex shapes. Therefore, we conduct the ablation study of these augmentations together. As shown in Tab. 2, we apply different sampling ratios to both augmentations (e.g., experiments id 1, 2, 3) and also exclude them in experiment 4. Boolean difference augmentation largely improves the metric (comparing experiment pair 2, 4 or 1, 3). Note that we use 60%/40% instead of 50%/50% because the later one has more instability (Sec. 5.2). The possible reason is visualized in Fig. 3: the lack of boolean augmentation in training data causes experiment 2 to show structural failure on concave shapes.

Table 2: Ablation studies over boolean difference and wireframe augmentations. The height-field (hf) is applied independently to each surface with prob. 0.5.

| | dataset | | | GSO | | | ABO | | |
|---|---|---|---|---|---|---|---|---|---|
| id | hf-only def. 40% | boolean def. 40% | wireframe def. 20% | PSNR↑ | SSIM↑ | LPIPS↓ | PSNR↑ | SSIM↑ | LPIPS↓ |
| 1 | default | default | default | **30.78** | **0.957** | **0.036** | **28.82** | **0.934** | **0.065** |
| 2 | 60.0% | 40.0% | 0% | 30.75 | 0.957 | 0.036 | 28.76 | 0.931 | 0.066 |
| 3 | 66.6% | 0% | 33.3% | 29.82 | 0.948 | 0.042 | 27.79 | 0.923 | 0.075 |
| 4 | 100.0% | 0% | 0% | 29.88 | 0.949 | 0.042 | 27.39 | 0.919 | 0.077 |

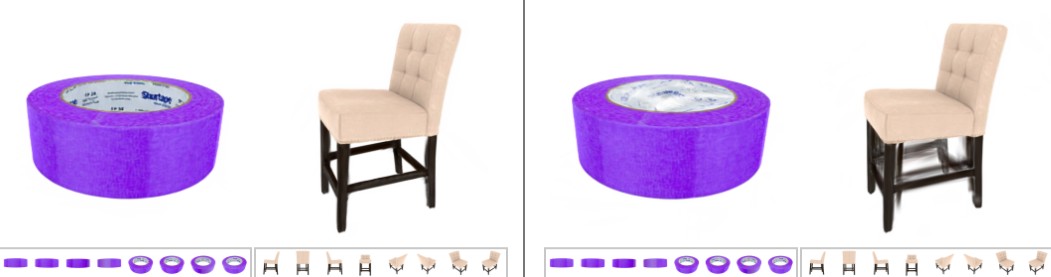

Figure 3: Qualitative results generated by *LRM-Zero* trained on *Zeroverse* with (left two) and without boolean difference augmentation (right two). Right two *LRM-Zero*'s reconstruction results have structural failures on objects with concave shapes and complex structures.

The wireframe augmentation does not show significant improvements of the metric, but it increases the visual fidelity. As shown in Fig. 4, without wireframe augmentation in its training data, *LRM-Zero* fails to reconstruct objects with thin structures, e.g. chair and table legs, or rails.

**Height-field augmentation** Tab. 3 shows two experiments with and without height field augmentation. Both are trained on 120K objects consisting of 80K original compositional objects and 40K boolean augmentation objects. This setting is different from other ablation experiments in Tab. 2, because we had to synthesize and render objects with 0 height field probability, which do not exist in *Zeroverse*. We also uses the boolean-difference only augmentation to mitigate the effect of instability. These results reveal that height field augmentation can improve the results.

## 5.2 Training stability

As discussed in Sec. 5.1, adding augmentation substantially boosts *LRM-Zero*'s performance. However, it also makes *Zeroverse* more complex and thus introduces training instability in *LRM-Zero*. We explore various techniques to help stabilize the training from either the training side (i.e., decreasing perceptual loss weight, decreasing Guassian splatting scale clipping, decreasing view-angle threshold) or the data mixing ratio of augmentations (we found that height-filed augmentation does not introduce instability a lot thus kept it). The observations are summarized in Tab. 4. In general, we observe that shifting training hyperparameters from optimal would improve the stability. However, this would decrease the performance. Thus our final plan (as shown in experiment 6) is a more balanced augmentation mixing ratio, and only minimal change on the training side. More comprehensive experiments are in Appendix.

## 5.3 Generalization

We first validate the generalization of our *Zeroverse* by training a NeRF-based LRM model [49, 92] on it. NeRF-based model's architecture is different from GS-LRM. Also the 3D modeling is philosophically different: NeRF has a canonical space for Triplane (i.e., Eulerian representation) while Gaussian Splatting is pixel-aligned per-point prediction (i.e., Lagrangian representation).

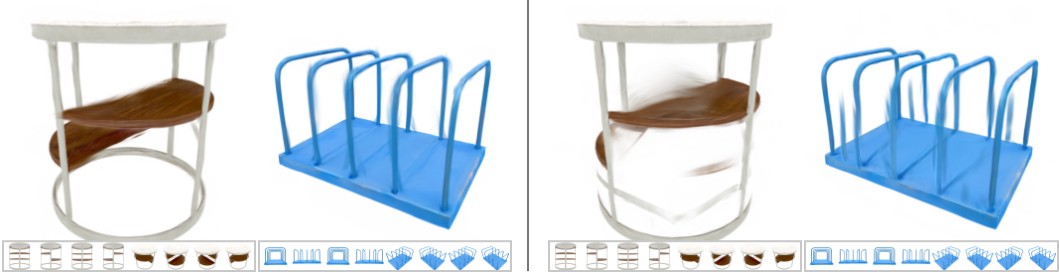

Figure 4: Qualitative results generated by *LRM-Zero* trained on default *Zeroverse* with (left two) and without wireframe augmentation (right two). Right two *LRM-Zero*'s reconstruction results have structural failures on objects with thin structures.

Table 3: Ablations studies on height-field augmentation.

| | dataset | | | Height Field | GSO | | | ABO | | |
|---|---|---|---|---|---|---|---|---|---|---|
| id | hf-only def. 40% | boolean def. 40% | wireframe def. 20% | HF probability def. 0.5 | PSNR↑ | SSIM↑ | LPIPS↓ | PSNR↑ | SSIM↑ | LPIPS↓ |
| 1 | 60.0% | 40.0% | 0% | default | **30.24** | **0.952** | **0.039** | **28.31** | **0.926** | **0.072** |
| 2 | | | | 0 | 29.22 | 0.941 | 0.045 | 27.70 | 0.916 | 0.076 |

Despite of these differences, the results in Tab. 5 are similar to what we observed in GS-LRM that *Zeroverse*-trained model is competitive to Objaverse-trained models.

Besides the standard benchmark GSO and ABO, we also evaluate our *LRM-Zero* on diverse datasets to show its generalization, such as realistic 3D objects in OpenIllumination [55] and OmniObject3D [95], cross-evaluation on Zeroverse and Objaverse, and the generative outputs by Instant3D [49] and One2345++ [57]. As these experiments are for generalization test, we use 8 randomly-sampled input for OpenIllumination, OmniObject3D, Objaverse, and Zeroverse. For Instant3D and One2345++, we use the default camera setup of the generative model's outputs, where Instant3D and One2345++ have 4 and 6 structural cameras, respectively. As shown in Tab. 6, our *LRM-Zero* is competitive. We visualize the Instant3D and One2345++ results in Fig. 5, where *LRM-Zero* still work for these truly novel generated images, showcasing that *LRM-Zero* can be used in the 3D generation pipeline.

# 6 Related works

3D reconstruction is an important task in 3D vision. As 3D data is usually hard to capture, 3D reconstruction gives the ability to get 3D model from other modalities (e.g., images). Traditional methods [66, 61, 56, 14] on 3D reconstruction focuses on the per-sample optimization, where the 3D

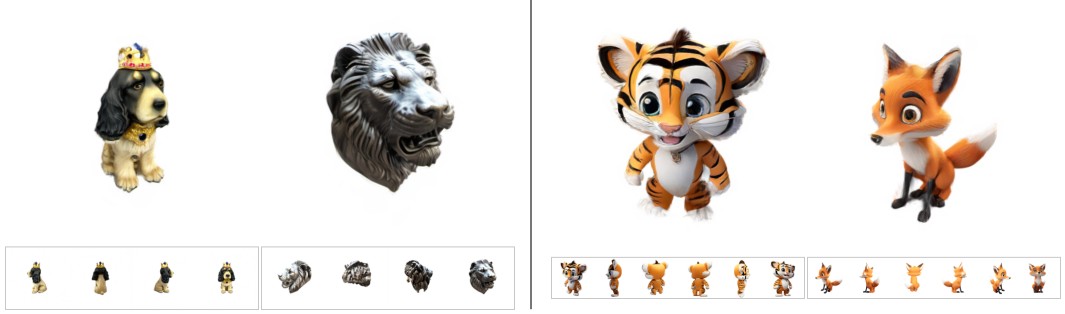

Figure 5: *LRM-Zero*'s qualitative results on Instant3D text-to-3D (left two) and One2345++ image-to-3D (right two) generated multi-view images.

Table 4: Illustrating the training stability issues when constructing the procedural *Zeroverse* dataset. The instability can be resolved either with training stabilizing techniques (e.g., reducing perceptual loss weight, Gaussian scale clipping, and view angle threshold), or with reducing the complexity of *Zeroverse*. 'failed' experiments are usually due to model divergence.

| | dataset | | | training | | | result |
|---|---|---|---|---|---|---|---|
| id | hf-only | boolean | wireframe | perceptual loss weight (default 0.5) | Gaussian scale clipping (default -1.2) | view angle threshold (default 60) | GSO PSNR, if finished |
| 1 | 100% | 0% | 0% | default | default | default | 29.54 |
| 2 | 20% | 80% | 0% | default | default | default | failed |
| 3 | | | | 0.2 | default | default | failed |
| 4 | 40% | 60% | 0% | 0.2 | default | default | failed |
| 5 | | | | 0.2 | -1.6 | 40 | 30.32 |
| 6 | 40% | 40% | 20% | 0.2 | default | default | 30.78 |

Table 5: NeRF-LRM-Zero performs competitively against NeRF-LRM-Objv.

| | GSO | | | ABO | | |
|---|---|---|---|---|---|---|
| | PSNR ↑ | SSIM ↑ | LPIPS ↓ | PSNR ↑ | SSIM ↑ | LPIPS ↓ |
| NeRF-LRM-Zero | 29.33 | **0.936** | **0.065** | 28.96 | 0.921 | 0.084 |
| NeRF-LRM-Objv | **29.72** | **0.936** | **0.064** | **30.79** | **0.932** | **0.071** |

shapes are parameterized and optimized by the rendering loss [61] or geometry loss [66, 25]. These optimization-based methods are usually slow and require adequate number of views (e.g., 100 views). Although methods are proposed [82, 30, 104, 62] to resolve these constraints for efficiency and view requirements [43, 64, 81], the speed is not largely improved.

Recent progresses advances this task with learning-based feed-forward methods [105, 94, 80, 63, 45, 50, 44]. Instead of optimization, these methods train a model from large-scale object [73, 106, 97, 23, 24] or scene [109, 71, 54] data to predict the shape directly. Besides the benefits of efficiency, these feed-forward methods can naturally support sparse-views as input (e.g., 4 to 12 input view images) because they learn data patterns from massive dataset. Some models can even go with extreme case of single-view reconstruction [105, 80, 41, 83], which needs to have data prior from realistic 3D data. Multi-view stereo methods [78, 103, 36, 87, 26, 88] are another family of feed-forward 3D reconstruction methods, but they cannot deal with sparse-view or single-view settings since they are based on local feature matching.

Synthetic data has been popular used in computer vision [38, 58, 32, 76], such as in segmentation [15, 21], object detection [42], image classification [39], deblurry [75], face analysis [93], etc. In 3D vision, synthetic data is widely used because of 3D data is harder to harvest, e.g., in depth estimation [4, 69], in optical flow [27, 60, 59], in finding multi-view correspondence [91], and for improving the 3D consistency of multi-view diffusion models [96]. Specifically to reconstruction, the

Table 6: Generalization of *LRM-Zero* to various evaluation datasets.

| | OpenIllumination | | | OmniObject3D | | | Objaverse-test | | | Zeroverse-test | | |
|---|---|---|---|---|---|---|---|---|---|---|---|---|
| | PSNR↑ | SSIM↑ | LPIPS↓ | PSNR↑ | SSIM↑ | LPIPS↓ | PSNR↑ | SSIM↑ | LPIPS↓ | PSNR↑ | SSIM↑ | LPIPS↓ |
| GS-LRM | 14.02 | **0.598** | 0.460 | **29.32** | **0.940** | **0.055** | **28.37** | **0.920** | **0.079** | 26.48 | 0.880 | 0.089 |
| *LRM-Zero* | **14.44** | 0.591 | **0.455** | 25.81 | 0.909 | 0.080 | 25.88 | 0.884 | 0.112 | **28.23** | **0.912** | **0.068** |

exploration of synthetic is mainly on specific categories, for example, for face [74], for human [33], constructions [47], or for evaluation [1, 31]. Some synthetic data are template-based [9, 34, 51, 102] and injecting human's knowledge about the semantic. Xu et al. [100, 101] and their subsequent works [52, 7, 53, 77, 110] have leveraged procedurally synthesized data for relighting, view synthesis, and various appearance acquisition and rendering tasks. However, these methods are designed for captures under controlled lighting conditions or objects with specific materials; additionally, their data is created on a relatively small scale. We revisit their procedural data generation workflow, extending it with additional data augmentation techniques and scaling it up to train large reconstruction models.

# 7 Limitations

In this paper, we mainly focus on providing a proof of concept on using synthetic to tackle one of the key problems in 3D vision: 3D reconstructions, and here are part of the limitations.

**Scalability.** The scalability of such synthetic-based method is still under investigation. We have done some initial exploration and the results can be found in Appendix. From these early experiments, it seems that the convergence property and optimal training hyperpameters might be different from the standard experimental setup with real data. The scaling-up exploration would naturally involve more resources (mainly computing resources, i.e., GPU hours) which is beyond our affordability.

Also, the community also lacks a study over the scalability of reconstruction models over 'real' data. Objaverse-XL [24] brings 10 more data over Objaverse but the data is much nosier, has different formats, contains a large portion without textures, and the legal concerns are not fully resolved. All these issues exposes challenges in understanding the scalability of the feed-forward reconstruction method.

**Semantics.** The synthetic data created in the way of Sec. 3 lacks of semantics (e.g., the data distribution is not supposed to match the real 3D world distribution). Thus this data might not be suitable to learn semantical-rich tasks. For the simplest example, *Zeroverse* is hard to train single-view reconstructions as shown in MCC [94], Shap-E [45], LRM [41], etc, which learn semantic from Objaverse [23], MvImgNet [106], and Co3D [73]. At the same time, we can complete single-view reconstruction by chaining with multi-view generator [57, 89] as shown in [49], relying on the semantical understanding of multi-view generation. The exact boundary of semantic tasks and intrinsic tasks in 3D vision is still under debate.

# 8 Broader Impacts

The broader impacts of this work are overall positive. First, the proof of concept in using synthesized data would largely reduce the bias inside the real dataset. As the model has weak inductive bias (i.e., through the use of the pure-transformer architecture), the potential semantical bias is mostly from the data. Second, the 3D data are usually having license concerns, where the synthesized data can help resolve. Third, as the 3D reconstruction can be potentially learned from synthetic data without real-world semantic information, we can possibly separate the 3D generation into two problems: generation and reconstruction. The reconstruction is mostly a semantics-free task.

On the other hands, this work can potentially largely lowers the bar of 3D reconstructions, for which data is the main blocker previously. The accessible 3D generation (when chaining with generative models as shown in [49]) and 3D reconstruction ability may introduce legal concerns on 3D licensing and moral concerns on 3D identities.

# 9 Conclusion

We introduced the *LRM-Zero* and its training data *Zeroverse*. *Zeroverse* is constructed with procedural synthesizing, where primitive shapes are composited, textured, and then augmented. We found the LRM model trained with *Zeroverse* can be competitive with Objaverse-trained LRMs, thus illustrating a promising direction of using synthetic data in 3D reconstruction research. We released our data creation code, and hope that it can help future research.

## Acknowledgments

We thank Kalyan Sunkavalli, Nathan Carr, Milos Hasan, Yang Zhou, and Jimei Yang for their support on this project.

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

# A    Appendix summary

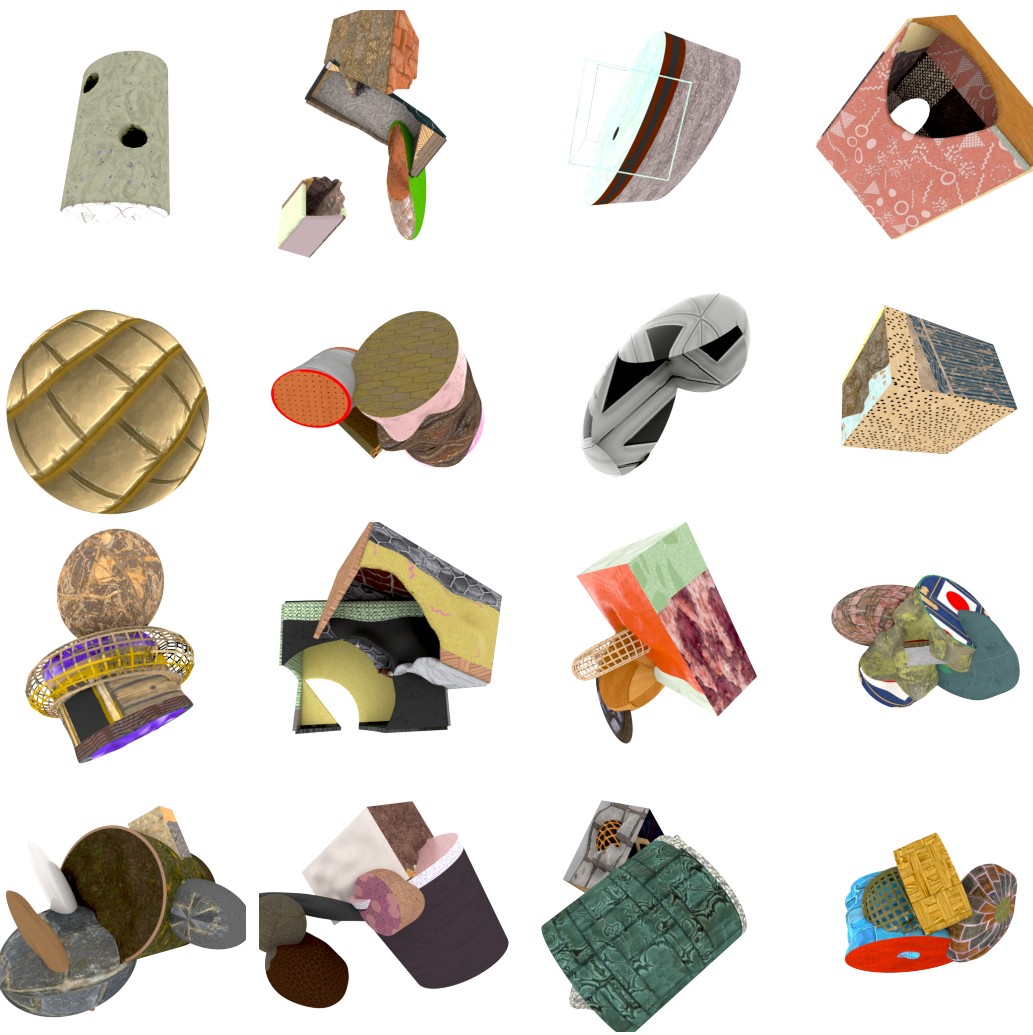

Figure 6: Uniformly sampled objects from *Zeroverse* to visualize its data distribution.

In Appendix, we provide more visualization results, more analysis, and more implementation details of our paper. Also note that both *LRM-Zero* and GS-LRM (when involving the results) refer to the final model (i.e., with respect to their training data and training parameters) instead of just the model architecture. We also use GS-LRM to refer to the model architecture as well for simplicity.

Table 7: Quantitative results comparing *LRM-Zero* with GS-LRM [107] under the 4-input-view setting. We use GSO [28] and ABO [18] evaluation datasets and PSNR, SSIM, and LPIPS [108] metrics. *LRM-Zero* demonstrates competitive performance against GS-LRM.

|  |  |  | GSO | | | ABO | | |
|---|---|---|---|---|---|---|---|---|
|  |  |  | PSNR ↑ | SSIM ↑ | LPIPS ↓ | PSNR ↑ | SSIM ↑ | LPIPS ↓ |
| 4 input views | Res-512 | GS-LRM | **30.52** | **0.952** | **0.050** | **29.09** | **0.925** | **0.085** |
|  |  | *LRM-Zero* | 28.49 | 0.937 | 0.063 | 25.40 | 0.893 | 0.115 |
|  | Res-256 | GS-LRM | **29.59** | **0.944** | **0.050** | **28.92** | **0.926** | **0.074** |
|  |  | *LRM-Zero* | 27.78 | 0.927 | 0.062 | 25.41 | 0.886 | 0.106 |

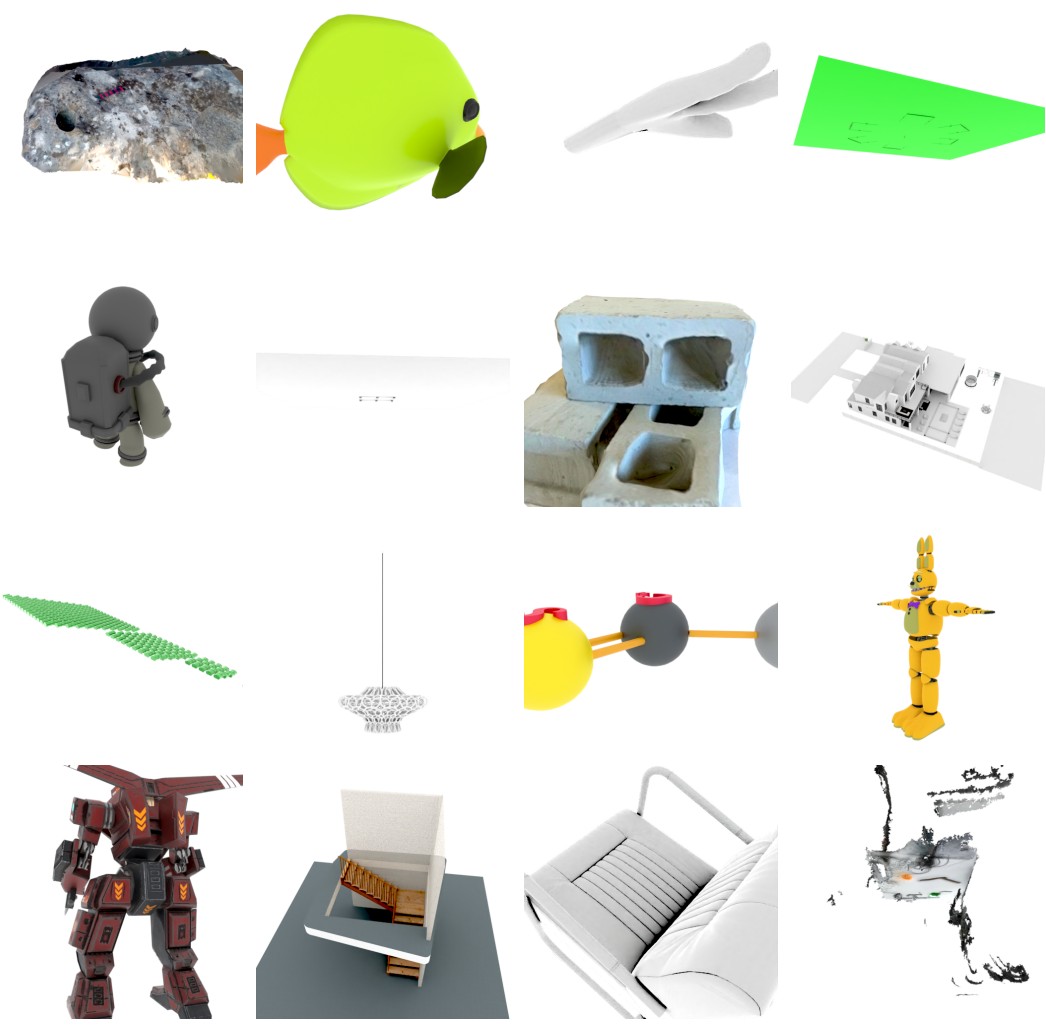

Figure 7: Uniformly sampled objects from Objavserse [23] to visualize its data distribution.

## B  Uniformly-sampled data visualization

In Fig 1, we visualize a curated set of the two dataset *Zeroverse* and Objaverse [23]. We here present uniformly random samples of the two dataset for better understandings of the data distribution *Zeroverse* is in Fig. 6 and Objaverse is in Fig. 7. Our *Zeroverse* is a random procedural data thus the datasest is quite diverse but do not have any semantic. The Objaverse dataset is sourced from Sketchfab [1]. The Objaverse dataset contains more semantic meaning (e.g., the transformers in the bottom left, some humanoids, furniture, and structure of house). There are also some shapes without explicit semantic meaning (e.g., the objects at position row-1-column-1, row-1-column3, row-3-column-1). Comparing the two datasets visually, usually the *Zeroverse* data is more complex and has higher-frequency textures than the Objaverse dataset in average. However, there are some data in Objaverse have more fine-grained small structures (e.g., the lamp at row-3-column-2) and more overall details (e.g., the transformer at bottom left), which currently the *Zeroverse* can not achieve. We think that these data difference contributed to the gap of the metric-wise results, but might be mitigated with improved procedural process. The visual difference is not significant as shown in Fig. 1. More visualization of *LRM-Zero* can be found at our website https://desaixie.github.io/lrm-zero/.

---

[1] https://sketchfab.com/

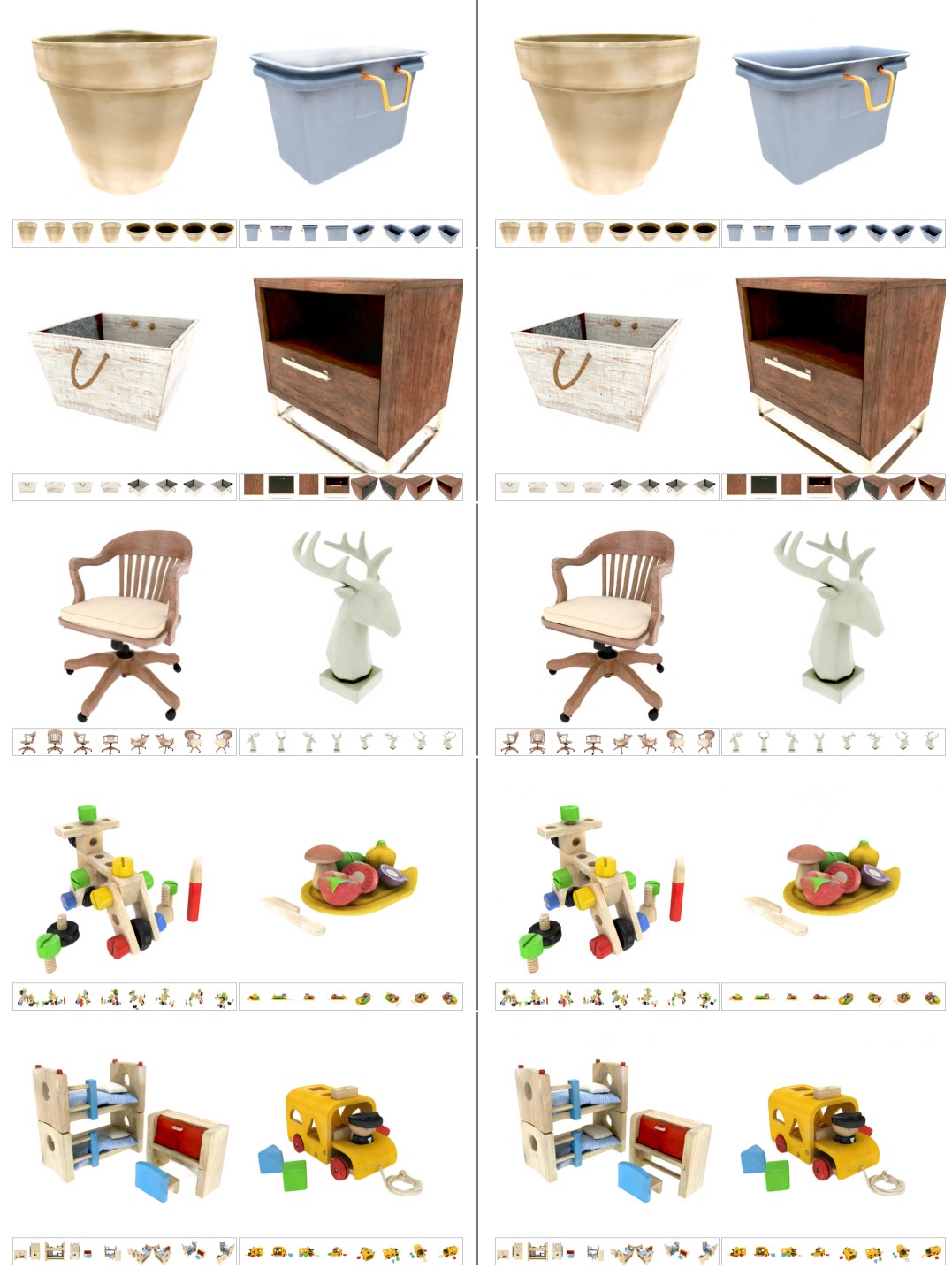

Figure 8: Qualitative comparison of *LRM-Zero* (left two columns) and GS-LRM (right two columns). When there is invisible region in the input views (first row), *LRM-Zero* produces poor reconstruction results. When the input views have good coverage (second row to fifth row), *LRM-Zero* performs similarly well as GS-LRM.

As shown in Tab. 1, *LRM-Zero* performs worse than GS-LRM on average. In Fig. 8, we show some qualitative comparisons of the two model's reconstruction results. Although we provide 8 input views, for the first row in Fig. 8, there are still invisible region in the input view. This leads to *LRM-Zero*'s

poor reconstruction results on them, as *LRM-Zero* does not learn sufficient semantics of real-world objects like GS-LRM. When there is sufficient coverage in the input views, as in the second row in Fig. 8, we can see that *LRM-Zero* and GS-LRM produces similar reconstruction results. In the third row to the fifth row, both *LRM-Zero* and GS-LRM performs similarly well on objects with complex shape.

On the other side, there is an extension of Objaverse named ObjaverseXL [24], which contains 10M 3D data from the Internet. We did not show it here and did not use it in our exploration for now. The quality of ObjavereXL is worse than Objaverse, and the data format (untextured shapes, point clouds) does not meet our data requirements. ObjaverseXL contains four subsets, 1. Sketchfab, which is the same to Objaverse, 2. Smithsonian [2], only about 2K data in this subset. 3. Thingiverse [3], with untextured mesh data. 4. Github, which we have not got license clearance on downloading all of the data. For these reasons, we did not use ObjaverseXL in our initial exploration.

## C  Early Exploration on Scalability of *LRM-Zero*

We get mixed results from our scalability experiments on training steps, model size, and data size. We hypothesize that training convergence is the key to *LRM-Zero*'s performance. For many experiments in Tab. 8, the model underperforms its counterpart due to undertraining and lack of convergence. We suspect that the complexity of *Zeroverse* and our modified, lowered perceptual loss scale (discussed in Sec. 5.2), and the lowered learning rate for 2x and 3x model sizes contribute to the limited training convergence of *LRM-Zero*.

Table 8: *LRM-Zero*'s scaling experiment results.

| | scaling | | | GSO | | | ABO | | |
|---|---|---|---|---|---|---|---|---|---|
| id | Training Steps def. 1x, 80K | Model Size def. 1x, 300M | Data Size def. 1x, 400K | PSNR ↑ | SSIM ↑ | LPIPS ↓ | PSNR ↑ | SSIM ↑ | LPIPS ↓ |
| 1 | 1x | 1x | 1x | 30.78 | 0.957 | 0.036 | 28.82 | 0.934 | 0.065 |
| 2 | 2x | 1x | 1x | 31.47 | 0.962 | 0.032 | 29.33 | 0.938 | 0.061 |
| 3 | 3x | 1x | 1x | 31.11 | 0.960 | 0.035 | 29.18 | 0.937 | 0.063 |
| 4 | 1x | 2x | 1x | 30.19 | 0.952 | 0.041 | 28.43 | 0.927 | 0.071 |
| 5 | 1x | 3x | 1x | 30.34 | 0.954 | 0.040 | 28.58 | 0.929 | 0.069 |
| 6 | 2x | 2x | 1x | 30.00 | 0.949 | 0.042 | 28.25 | 0.925 | 0.073 |
| 7 | 2x | 2x | 2x | 30.56 | 0.955 | 0.038 | 28.84 | 0.931 | 0.068 |
| 8 | 2x | 3x | 10x | 30.10 | 0.951 | 0.042 | 28.28 | 0.926 | 0.073 |
| 9 | 2x | 1x | 4x | 31.15 | 0.960 | 0.034 | 29.02 | 0.935 | 0.064 |
| 10 | 2x | 1x | 20x | 31.08 | 0.960 | 0.034 | 28.95 | 0.936 | 0.063 |
| 11 | 3x | 1x | 20x | **31.51** | **0.963** | **0.031** | **29.41** | **0.940** | **0.060** |

**Training steps**  We discover that both GS-LRM [107] (experiment 2 in Tab. 9) and *LRM-Zero* (experiment 2 in Tab. 8) benefit from training on 2x (160K) steps, which was not explored in [107]. The gap between *LRM-Zero* and GS-LRM at 2x training steps is larger than the gap at 1x training steps. This shows that GS-LRM models, especially *LRM-Zero*, are undertrained at 1x (80K) training steps and aligns with our hypothesis that training convergence is the key.

At 1x (400K) training data size, Experiment 3 in Tab. 8, with 3x training steps, performs worse than experiment 2 in Tab. 8, indicating that the optimal training step for 1x model size and 1x *Zeroverse* data is somewhere between 2x and 3x. At 20x (8M) data size, 3x training steps (experiment 11) outperforms 2x training steps (experiment 10), indicating that more training steps is needed to

---

[2]https://3d.si.edu/
[3]https://www.thingiverse.com/

Table 9: GS-LRM's scaling experiment results.

| | scaling | | | GSO | | | ABO | | |
|---|---|---|---|---|---|---|---|---|---|
| id | Training Steps def. 1x, 80K | Model Size def. 1x, 300M | Data Size def. 2x, 800K | PSNR↑ | SSIM↑ | LPIPS↓ | PSNR↑ | SSIM↑ | LPIPS↓ |
| 1 | 1x | 1x | 2x | 31.90 | 0.966 | 0.030 | 30.66 | 0.949 | 0.055 |
| 2 | 2x | 1x | 2x | **33.12** | **0.973** | **0.024** | **31.75** | **0.957** | **0.047** |

converge well on more training data. This again aligns with our hypothesis about the importance of training convergence.

**Model size** We experiment with 2x (700M parameters) and 3x (1B parameters) model sizes with a lowered learning rate of 3-e4 instead of the default 4e-4. At both 1x (experiment 4, 5) and 2x training steps (experiment 6, 7, 8), the larger models underperform compared their 1x model size counterparts (experiment 2, 3). We suspect that this is due to the fact that they are trained with a reduced learning rate of 3e-4 instead of the default 4e-4 for training stability purpose, which limits the models' training convergence. Also, we might need to tune more hyperparameters for the 2x and 3x model sizes to see their benefit. Due to the limited computation resources, we leave the exploration of larger model sizes as future work.

**Data size** At 2x training steps with 1x model size, increasing data size in experiments 9 and 10 does not help model's performance compared to experiment 2. At 3x training steps with 1x model size, experiment 11 with 20x training data outperforms experiment 3, which overfits on 1x training data. By training on 8M *Zeroverse* objects with sufficient training steps, Experiment 11 is also our overall best model. We also observe that experiment 11 performs similarly as experiment 2, which uses only 400K training data. We hypothesize this is because the benefit of 20x data is limited by the capacity of the 1x model and that training a larger model on 20x data is promising. Due to the limited computing resources, we leave this as future work.

## D   Zeroverse creation details

### D.1   Sampling and Compositions of Primitives

We randomly sampled the number of the primitives from 1-to-9, with an unnormalized probability density of [5, 5, 5, 5, 5, 4, 3, 2, 1] accordingly. This sampling strategy samples more shapes with lower number of primitives that can smooth the dataset and increase the LRM training stability. Also, the later augmentations (especially the wireframe conversion and the boolean difference) can possibly introduce more fractions and parts.

For sampled primitives, we will scales their size randomly through each axis of the shape. We first put the shape in a normalized frames (e.g., align each edge of the cube with one of the axis) and randomly sample the scaling factors of each axis independently.

For compositing the scaled primitives, we randomly sample the centers of each primitive in the bounding box $[-1, 1]^3$, then we randomly sample a rotation of the shape (with three degrees of freedom). The compsition is done by simply putting all primitives together to a single shape. We do not consider the connectivity of the synthesized objects. See Fig. 6 for an example of disjoint shape (e.g., the top right one).

### D.2   Augmentations

For the height-field augmentations, we constrain the maximal value of the height map to be proportional to the size of the face to avoid over-displacement. For more details, please refer to Xu et al. [100] and their released codebase.

For the boolean difference augmentation and wireframe conversion augmentation, we use the function from Blender [4]. In details, we use Blender's boolean modifier and solidify modifier to augment the initial shape. We first add a random primitive shape with random size and at a random vertex on the initial shape. Then, we apply the boolean modifier, where the new primitive shape, treated as a cutter, is subtracted from the initial shape. We do not use the torus for the subtraction as it might reduce training stability. Finally, we apply the solidify modifier to the cut shape to add thickness to it. The inside faces of the resulting cut shape will have the same texture as the outside faces.

We use Blender's wireframe modifier and subdivision modifier to create the wireframe of a primitive shape. We also add randomness to the thickness of the wireframe. The wireframes make up the thin structures that are missing in the initial shapes (Sec. 3.1) and add diversity to *Zeroverse*.

The augmentation is applied randomly with a given probability. We do not apply 'boolean difference' and 'wireframe' augmentations at the same time. This will breaks the independence assumption of different augmentations, but would avoid ultra-complex shapes, which improve training stability of the reconstruction model. In our implementation, we by default take $0.4$ probability of the 'boolean difference' augmentation, $0.2$ of the 'wireframe', and $0.4$ of not applying both (as we want disjoint distribution for 'boolean difference' and 'wireframe' augmentations). The 'height-map' augmentation is applied independently to the above two shape augmentations, and always set at $0.5$ independently for each surface. The results of other configuration can be found in the stability section (Sec. 5.2).

## E    Data synthesis distributed implementation details

In order to synthesize the massive amount of data, we tackle the synthesis and rendering of 8M shapes in *Zeroverse* with job parallelization. We run the independent shape synthesis and rendering jobs in parallel on 400 CPU nodes with a total of 38,400 CPU cores. The whole process takes 88 hours, where 5% of the time is spent on shape synthesis and 95% of the time is spent on rendering. Despite the relatively large number of CPU cores, the cost is negligible comparing with the training experiments cost on GPU. The training experiments in the main paper are mostly carried on a subset of 400K of the data. The early exploration of scalability uses the full dataset.

To avoid duplication in job parallelism, we first assign unique uuids and corresponding seeds for each shape to synthesize. Then, given the uuid, the seed, and the seed-induced fixed set of parameters, the synthesis and rendering jobs of each shape is independent from each other. In order to avoid exceeding local disk storage, we regularly upload the synthesized shapes and the rendered images to remote storage (e.g., AWS s3 in our experiment) and free the memory of their local copies. On each CPU node, we run multiple shape synthesis and rendering jobs in parallel to maximize the CPU utilization.

## F    More training stability results

As discussed in Sec. 5.1, adding augmentation substantially boosts *LRM-Zero*'s performance. However, it also makes *Zeroverse* much more complex and thus *LRM-Zero*'s training unstable, as shown in experiment 2 and 5 in Tab. 10. We explore various techniques to help stabilize the training. First, we adjust the perceptual loss weight. Compared to GS-LRM [107] trained on Objaverse [23] and *LRM-Zero* trained on no-augmentation *Zeroverse*, boolean augmented *Zeroverse* objects have high perceptual loss magnitudes that causes the excessive gradient norm. In experiment 2, we observe unusual, excessive gradient norm values in the range of 2-5. By reducing perceptual loss weight from 0.5 to 0.2 in experiment 3, the gradient norm values drop to the reasonable range of 0-1. However, experiment 3 still failed due to gradient norm explosion later.

Suspecting that boolean-augmented objects added too much dataset complexity, we reduced their ratio while increased the ratio no-augmentation objects in experiment 5 but still ended up with gradient norm explosion. Then, while keeping the same training dataset, we experimented with two techniques to further stabilize the training, i.e. reducing the Gaussian's scale clipping and the view angle threshold between the sampled views, in experiment 6, 7, and 8. Both techniques turn out to allow model to train stably for the full training steps. However, we notice that they also reduce the model's training set convergence and testing set performance.

---

[4] https://docs.blender.org/manual/en/latest/copyright.html

Table 10: Illustrating the training stability issues when constructing the procedural *Zeroverse* dataset. The instability can be resolved either with training stabilizing techniques (e.g., reducing perceptual loss weight, Gaussian scale clipping, and view angle threshold), or with reducing the complexity of *Zeroverse*. 'failed' experiments are usually due to model divergence.

| | dataset | | | training | | | result |
|---|---|---|---|---|---|---|---|
| id | hf-only | boolean | wireframe | perceptual loss weight (default 0.5) | Gaussian scale clipping (default -1.2) | view angle threshold (default 60) | GSO PSNR, if finished |
| 1 | 100% | 0% | 0% | default | default | default | 29.54 |
| 2 | 20% | 80% | 0% | default | default | default | failed |
| 3 | | | | 0.2 | default | default | failed |
| 4 | | | | 0 | default | default | failed |
| 5 | 40% | 60% | 0% | 0.2 | default | default | failed |
| 6 | | | | 0.2 | -1.6 | 40 | 30.32 |
| 7 | | | | 0.2 | -1.6 | default | 30.42 |
| 8 | | | | 0.2 | default | 40 | 30.85 |
| 9 | 92% | 8% | 0% | 0.2 | -1.6 | 40 | 30.14 |
| 10 | | | | 0.2 | -1.6 | default | 30.46 |
| 11 | | | | 0.2 | default | default | 30.86 |
| 12 | 40% | 40% | 20% | 0.2 | default | default | 30.78 |
| 13 | 85% | 10% | 5% | default | default | default | 30.62 |

We shift the data distribution of *Zeroverse* to include more easy data and less complex boolean-augmented shapes. In experiment 9, 10, and 11, we find that training stabilizing techniques constraints the model's training set convergence and testing set performance. Experiment 11 further shows that by reducing the ratio of boolean-augmented complex shapes, we can stably train *LRM-Zero*. In our final version of *Zeroverse*, we aim to reduce the dataset complexity in order to achieve stable training from the data distribution side to avoid the convergence constraints that Gaussian's scale clipping or view angle threshold entails. In our empirical observation of the training data, we notice that adding boolean augmentation to objects with many basic shapes can be over complex. In experiment 12, which has the same data distribution as our final version of *Zeroverse*, we adopt a more even, smooth distribution between no-aug, boolean, and wireframe augmented objects and an easier basic-shape-number distribution to favor less basic shapes per object. It performs similarly as experiment 8 and 11. Additionally, in experiment 13, when reducing the ratio of data with boolean difference augmentation, we do not have training instability issues with the default training hyperparameters from GS-LRM, including the 0.5 perceptual loss weight.

# G  Additional Experiments

We conduct additional experiments to seek more explanations on the performance gap between *LRM-Zero* and GS-LRM. We perform an experiment on GS-LRM, training it on a randomly sampled subset of 200K Objaverse objects. As shown in Tab. 11, GS-LRM's performance only drops by 0.1 PSNR on GSO. We conduct joint training on both Objaverse and *Zeroverse* and compare it to training only on one of the two datasets in Tab. 12. Our result shows that training on both Objaverse and Zeroverse performs better than Zeroverse only, but worse than Objaverse only. Both of these experiments likely indicate that for the single-object reconstruction task, the results start to saturate with about 200K realistic data. Since the size of Objaverse is adequate for the single object reconstruction task, and the advantages of Zeroverse are not exploited in this task. However, we believe that the advantages

Table 11: Scaling down GS-LRM's training data size. When training on only 200K instead of 800K Objaverse data, GS-LRM's performance drops by only 0.1 PSNR on GSO.

| | scaling | | | GSO | | | ABO | | |
|---|---|---|---|---|---|---|---|---|---|
| id | Training Steps def. 1x, 80K | Model Size def. 1x, 300M | Data Size def. 2x, 800K | PSNR ↑ | SSIM ↑ | LPIPS ↓ | PSNR ↑ | SSIM ↑ | LPIPS ↓ |
| 1 | 1x | 1x | 2x | **29.59** | **0.944** | **0.050** | **28.92** | **0.926** | **0.074** |
| 2 | 1x | 1x | 0.5x | 29.42 | 0.942 | 0.052 | 28.75 | 0.924 | 0.075 |

Table 12: *LRM-Zero* vs. *LRM-Zero-Obja* vs. GS-LRM at the 8-input-view, 256 resolution setting. Z means Zeroverse and O means Objaverse. The *LRM-Zero* (first row) and GS-LRM (second row) results are from experiment 9 in Tab. 8 and experiment 2 in Tab. 9. The *LRM-Zero-Obja* result (third row) is obtained by training on 800K *Zeroverse* data and 800K Objaverse data. While *LRM-Zero-Obja* outperforms *LRM-Zero*, it underperforms GS-LRM.

| | scaling | | | GSO | | | ABO | | |
|---|---|---|---|---|---|---|---|---|---|
| data | Training Steps def. 1x, 80K | Model Size def. 1x, 300M | Data Size def. 1x, 400K | PSNR ↑ | SSIM ↑ | LPIPS ↓ | PSNR ↑ | SSIM ↑ | LPIPS ↓ |
| Z | 2x | 1x | 4x | 31.15 | 0.960 | 0.034 | 29.02 | 0.935 | 0.064 |
| O | 2x | 1x | 2x | **33.12** | **0.973** | **0.024** | **31.75** | **0.957** | **0.047** |
| Z&O | 2x | 1x | 4x | 32.11 | 0.968 | 0.027 | 30.70 | 0.950 | 0.052 |

of *Zeroverse*, i.e. the data size, texture quality, controllability are more valuable when extended to other tasks, such as scene reconstruction and relighting, where data is scarse.

