# OpenReview forum: "LRM-Zero: Training Large Reconstruction Models with Synthesized Data"
_NeurIPS.cc/2024/Conference — NeurIPS 2024 poster_

### Official Review · Reviewer_3hvP · 2024-07-12

**Soundness:** 3
**Presentation:** 3
**Contribution:** 3
**Rating:** 6
**Confidence:** 4

**Summary:**

This paper proposes Zeroverse, a new dataset that is entirely synthesized for training large feed-forward 3D reconstruction models. Based on Zeroverse, LRM-Zero is trained with the network structure of GS-LRM. LRM-Zero achieves comparable performance as GS-LRM, which is trained on Objaverse. The idea of using entirely synthesized data to train large reconstruction models is meaningful for the community. However, the performance of LRM-zero is usually worse than GS-LRM, e.g., OmniObject3D. Zeroverse also introduces stability issue for the training of LRM-Zero. It is unclear whether other methods need to carefully adjust the settings of Zeroverse for training stability.

**Strengths:**

The paper is overall clearly written.

The idea of using entirely synthesized data to train large reconstruction models is interesting and meaningful for the community. The commonly used Objaverse requires lots of efforts to collected and crafted by humans, and is known to have many low-quality 3D models that need to be cleaned in many recent works.

With Zeroverse, both the NeRF model (NeRF-LRM-Zero) and 3DGS model (LRM-Zero) can achieve comparable performance as the corresponding models trained on Objaverse.

Ablation study shows the effectiveness of different augmentations.

**Weaknesses:**

On OmniObject3D, which is not used for training by both GS-LRM and LRM-zero, the performance of LRM-zero is worse than GS-LRM, especially in PSNR.

As discussed in Sec. 5.2, Zeroverse may introduce stability issue for the training of 3D reconstruction models. The authors carefully tune the settings and hyperparameters to stably train LRM-Zero. But it is unclear whether other methods need to further adjust the settings for training stability.

**Questions:**

Related works section is not adequate. Before NeRF is used for 3D reconstruction, 3D reconstruction is studied for decades already and multi-view stereo is the main family for 3D reconstruction. These methods [1*-6*], including some recent deep-learning based methods, should be cited and discussed since they are robust, feed-forward, able to scale to very large outdoor scenes and can handle sparse-view settings. But they cannot deal with very sparse setting or single-view setting since they are based on local feature matching, instead of global semantics like LRM.

L132: In public release version, seems the texture dataset is different from that is used in the paper. Will the performance become worse with the released texture dataset?

Would it be interesting to finetune LRM-Zero on a small set of high-quality captured scenes to learn semantics and improve the performance?

[1*] Schönberger et al. Pixelwise view selection for unstructured multi-view stereo. ECCV 2016.

[2*] Yao et al. Mvsnet: Depth inference for unstructured multi-view stereo. ECCV 2018.

[3*] Gu et al. Cascade cost volume for high-resolution multi-view stereo and stereo matching. CVPR 2020.

[4*] Wang et al. PatchmatchNet: Learned Multi-View Patchmatch Stereo. CVPR 2021.

[5*] Ding et al. TransMVSNet: Global Context-aware Multi-view Stereo Network with Transformers. CVPR 2022.

[6*] Wang et al. IterMVS: Iterative Probability Estimation for Efficient Multi-View Stereo. CVPR 2022.

**Limitations:**

Limitations are discussed in Sec. B of supplementary. Societal impacts are discussed in Sec. C of supplementary.

---

> ### Author Rebuttal · Authors · 2024-08-07
>
> We thank the reviewer for appreciating our writing clarity, the novelty and the value of our method, the generalization of Zeroverse for training both NeRF and 3DGS-based LRM models, and the effectiveness of our ablation study. We will respond to the reviewer’s comments below:
>
> 1. **Results on OmniObject3D**: As shown in Tab. 6, LRM-Zero has a 3.51 PSNR gap to GS-LRM on OmniObject3D. We believe that this is because OmniObject3D objects are more realistic (real captures) and have more diverse semantics than other testing datasets, e.g. GSO and ABO. Since LRM-Zero does not learn real-world object semantics from its training data Zeroverse, it thus performs worse on OmniObject3D than GS-LRM, which learns semantics from its training data Objaverse.
> 2. **Training instability**: We have identified that having too much boolean difference augmented objects in the training data is the main cause of our training instability issues. As shown in experiment 13 in Tab. 3 in the rebuttal pdf, when reducing the ratio of boolean difference augmentation, we do not need to change any training hyperparameters from GS-LRM, including the 0.5 perceptual loss weight. Thus, when extending LRM-Zero/Zeroverse to other tasks, if there is no substantial change in the Zeroverse data synthesis pipeline, there should not be training instability issues as long as the ratio of boolean difference augmentation is kept low. If not, our training stability experiences still provide valuable guidance: avoid making the synthetic object too complex, e.g. with boolean difference augmentation.
> 3. **Related works on multi-view stereo**: Thanks for curating the list of relevant works on MVS. We will include them in the revision.

---

> ### Comment · Reviewer_3hvP · 2024-08-12
>
> Thanks for the authors' reply. Some of my questions are addressed. However, I think my following questions are not answered:
>
> (1) L132: In public release version, seems the texture dataset is different from that is used in the paper. Will the performance become worse with the released texture dataset?
>
> (2) Would it be interesting to finetune LRM-Zero on a small set of high-quality captured scenes to learn semantics and improve the performance?
>
> For question (1), authors can answer it if they have any results on the public texture dataset. For question (2), though authors do not answer explicitly, I think the third row in Tab.1 of the rebuttal pdf shows that such finetuning on the Objaverse dataset with semantics may not help to improve performance on object reconstruction.

---

> > ### Author Response · Authors · 2024-08-13
> >
> > Thanks for the reply and below are our response:
> >
> > 1. **Texture dataset**: We have not tested with the open material dataset due to the limited time and computation cost (need to re-render all objects with new materials and run the training). In our observations, LRM relies more on low-level correspondence instead of high-level semantics for reconstruction, and therefore switching to the public material dataset is not expected to have a big impact on the quality. We will also release our original training shapes and images upon approval for better reproducibility.
> > 2. **Finetune LRM-Zero on data with semantics**: Thanks for the suggestion. We did a preliminary experiment where we fine-tune the Zeroverse model with Objaverse data. We observe that the fine-tuning leads to better results (PSNR increased from 31.2 to 32.3 on GSO) than Zeroverse-only training but can not beat training on Objaverse data. It’s possible that a more careful filtering to better bridge the distribution gap would lead to better results, and a more careful tuning of the fine-tuning parameters such as learning rate would also further boost the results.
> >    We also want to clarify the results in rebuttal Table 1: we do Zeroverse training in row 1, Objaverse training in row 2, and Zeroverse+Objaverse joint training in row 3 (i.e., not fine-tuning). We found that join training (row 3\) can improve from Zeroverse-only training (row 1\) results. However, since joint-training merges two data distributions, which is more challenging than training on either Zeroverse or Objaverse, it requires tuning our model configurations (such as a larger network capacity) and training hyperparameters that we did not perform. This explains why joint training (row 3\) performs worse than training only on Objaverse (row 2).

---

> ### Comment · Reviewer_3hvP · 2024-08-14
>
> Thanks for the response. It would be nice to add the results on public texture dataset in the final version to help the readers understand the performance difference.

---

### Official Review · Reviewer_xM78 · 2024-07-19

**Soundness:** 3
**Presentation:** 3
**Contribution:** 2
**Rating:** 5
**Confidence:** 4

**Summary:**

This paper explores an unusual route of training a large-scale 3D reconstruction model using synthetic data. It demonstrates that high-quality reconstruction can be achieved solely with synthetic procedural data, bypassing the need for real, hand-crafted 3D models, which are challenging to collect. The paper, trains two reconstruction models, one with objaverse dataset and the other with synthetic dataset which the paper proposes (Zeroverse). Test results of these models on ABO and Google Scanned Dataset shows that competitative reconstruction quality can be achieved by just synthetic data. With this the paper showcases that, global semantics of an object are not crucial for reconstruction. Consequently, similar reconstruction quality can be attained using complex geometric synthetic data with rich textures, even if they lack global semantics.

**Strengths:**

- **Novelty** - The use of synthetic data for reconstruction task is novel. The synthetic data generated by the method in this paper can also be used for data augmentation in other tasks.
- **Clarity** - The paper is well written with good attention to detail.
- **Results** - The approach has been appropriately validated on different datasets (Google Scanned and ABO).
- **Related Work** - Comprehensive releated works are covered in the paper.

**Weaknesses:**

- The paper lacks enough technical contributions. It seems that the entire paper is about ways to create synthetic data using predefined primitives and applying augmentations over different combinations of these primitives. After this once the data is prepared, an off-the-shelf gaussian splatting based reconstruction model is trained for performing experiments.
- From the results in Table 7 in supplementary, it seems that reconstruction quality significantly suffers (~2-3.5 PSNR) when the input views are very sparse (4 views) as compared to results with 8 input views as shown in Table 1. What is the reason behind this performance dip? Is it the object semantics?
- As this approach requires comparatively denser views (at least 8), I am concerned about its application in learning 3D priors, as generally 3D prior models work with extremely sparse views.
- The paper compares results by training models with same quantity of data for both real and synthetic datasets. It would be interesting to see if the reconstruction quality of the model trained with synthetic data improves or achieves comparable results when the dataset size is increased.

**Questions:**

Please refer to weakness.

**Limitations:**

Adequate limitations are discussed by the authors in Appendix.

---

> ### Author Rebuttal · Authors · 2024-08-07
>
> We thank the reviewer for appreciating the novelty and potential value for future works, the clarify of our writing, the comprehensiveness of our experiments and related works. We respond to the questions of the reviewer below:
>
> 1. **Technical contributions**: Building upon the prior work Xu et al. \[1\], we have added boolean difference and wireframe augmentations, which are crucial to the performance of LRM-Zero, and scaled up the data synthesis pipeline to synthesize and render up to 8M objects. Using the GS-LRM \[2\] model architecture, we also iterated on data and model design together to resolve training instability issues and achieve competitive performance with GS-LRM. We believe that such experiences are major contributions in the large scale training era. For this reason, we also share all the details about our experiences in the paper, which is valuable to the community, as recognized by reviewer **NfXg**.
> 2. **Sparse-view instead of dense-view setting**: This is a good question. LRM-Zero indeed has a larger performance gap to GS-LRM under the sparse-view reconstruction setting, as shown in Tab. 1 and 7\. This is because LRM-Zero does not learn real-world object semantics from Zeroverse. As a result, LRM-Zero cannot hallucinate the appearance and shape of an object where there is occlusion, which is common in the sparse-view setting. The sparse-view reconstruction task requires generation ability and knowledge of object semantics from the reconstruction model, which in turn requires training on a large dataset of realistic objects, e.g. Objaverse. LRM-Zero is more suitable for the dense-view reconstruction task. See the discussion of this limitation in Sec. B. Semantics in the Appendix.
> 3. **Application of a dense-view reconstructor**: As discussed above in 2., Zeroverse and LRM-Zero are not suitable for tasks that require generation ability and object semantics. To compensate for this, one can combine LRM-Zero with a generative model that possesses rich object semantics, e.g. the text-to-multi-view diffusion model from Instant3D \[1\]. Also, when working with extremely sparse input views, we can use a video diffusion model to obtain dense input views, and then do the reconstruction with LRM-Zero.
> 4. **Larger data size**: We show LRM-Zero trained on more Zeroverse data (2x, 4x, 10x, and 20x) in Tab. 8 in the Appendix. As discussed in Sec. E Data size in the Appendix, we do not observe visible gains in increasing the data size, even up to 20x. Furthermore, when increasing the data size without also increasing the training steps, the model cannot converge well and performance worse than when using less data (experiments 2, 9, and 10 in Tab. 8). We also perform an experiment on GS-LRM, training it on a randomly sampled subset of 200K Objaverse objects. As shown in Tab. 2 in the rebuttal pdf, GS-LRM’s performance only drops by 0.1 PSNR on GSO. This is likely because for the single-object reconstruction task, the results start to saturate with about 200K realistic data. However, we believe that the advantages of Zeroverse, i.e. the data size, texture quality, controllability are more valuable when extended to other tasks, such as scene reconstruction and relighting, where data is scarse (see Sec. B Beyond object-level reconstructions in Appendix).
>
> References
> \[1\] Jiahao Li, Hao Tan, Kai Zhang, Zexiang Xu, Fujun Luan, Yinghao Xu, Yicong Hong, Kalyan Sunkavalli, Greg Shakhnarovich, and Sai Bi. Instant3d: Fast text-to-3d with sparse-view generation and large reconstruction model. arXiv preprint arXiv:2311.06214, 2023
> \[2\] Kai Zhang, Sai Bi, Hao Tan, Yuanbo Xiangli, Nanxuan Zhao, Kalyan Sunkavalli, and Zexiang Xu. Gs-lrm: Large reconstruction model for 3d gaussian splatting, 2024\.

---

> > ### Comment · Reviewer_xM78 · 2024-08-12
> >
> > I thank the authors for the rebuttal. After reading the rebuttal and other comments by the reviewers I would like to keep my original rating.

---

### Official Review · Reviewer_CePT · 2024-07-19

**Soundness:** 4
**Presentation:** 4
**Contribution:** 3
**Rating:** 7
**Confidence:** 4

**Summary:**

This paper proposes a pure synthetic training dataset named Zeroverse which is composed of synthetic data generated by simple shape primitives and textures without any real-world semantics. With the Zeroverse dataset, the authors trained a GS-LRM 3D construction model called LRM-Zero and showed that LRM-Zero achieved comparable results to GS-LRM trained on real-world data. The extensive evaluations result to prove the feasibility of using pure synthetic data for 3D reconstruction as well as the effectiveness of the proposed data augmentation techniques.

**Strengths:**

- This paper is well-written and has a clear presentation, comprehensive evaluations, and insightful discussions.
- The proposed dataset, Zeroverse, has been shown to deliver great 3D reconstruction results in a sim-to-real generalization manner. This is a very valuable contribution to the community since it proposes a novel aspect of generating synthetic data for 3D reconstruction: the dataset doesn't have to contain real-world semantic objects at all to achieve good 3D reconstruction results. Also, I agree with the argument that 3D reconstruction relies heavily on local cues.
- The evaluation results are comprehensive and show that LRM-Zero achieves comparable results to GS-LRM on all the real-world dataset being evaluated.

**Weaknesses:**

- Although Zeroverse and LRM-Zero show very impressive results, it is unclear to me how to improve the result further. LRM-Zero still falls behind GS-LRM, though very closely, and the last bit may be on the semantics. I don't see a clear path to further improve Zeroverse, so the real-world applicability is questionable.
- Although the idea of generating a pure synthetic dataset for 3D reconstruction is novel, the dataset generation method and rendering method themselves are mainly taken from the prior work, *Deep image-based relighting from optimal sparse samples*. So there is not much novelty in the dataset generation algorithm.
- Training on Zeroverse seems to introduce more instability and one needs to spend more effort on the training tuning. Also, I don't quite understand why training on perfect synthetic data can introduce instability and it could mean that there is something did not work as expected on the data generation side.

**Questions:**

- It would be great if the authors could show a few representative examples where GS-LRM works much better than LRM-Zero, or vice versa. This way we can understand better where the gap is from.
- Does it help to use Zeroverse as pretraining and then finetuning on Objectverse? Or how about mixing them during training?

**Limitations:**

The authors have discussed the limitations in supplementary.

---

> ### Author Rebuttal · Authors · 2024-08-07
>
> We thank the reviewers for appreciating our writing quality, the value of the contribution of our work, and the comprehensiveness of our experiment results. We will address the questions of the reviewer below:
>
> 1. **Further improving LRM-Zero**: This is a good question. We have made many attempts to reduce the gap between LRM-Zero and GS-LRM, including adjusting the ratios of augmentation, training hyperparameters, and scaling up the model size, data size, and training steps. However, our best result reported in the paper still has a gap to GS-LRM, as shown in Tab. 1\. Among our attempts, increasing the data size was the most promising one, since we can synthesize unlimited amount of Zeroverse data, but it did not provide visible benefits, even at 2x or 3x model sizes, as shown in Tab. 8\. This leads to a conclusion that LRM models trained on Zeroverse has a gap to Objaverse on the single object reconstruction task, where Objaverse is sufficient. However, we believe that the advantages of Zeroverse, i.e. the data size, texture quality, controllability are more valuable when extended to other tasks, such as scene reconstruction and relighting, where data is scarce (see Sec. B Beyond object-level reconstructions in Appendix).
> 2. **Novelty of our method**: Building upon the prior work Xu et al. \[1\], we have added boolean difference and wireframe augmentations, which are crucial to the performance of LRM-Zero, and scaled up the data synthesis pipeline to synthesize and render up to 8M objects. We also iterated on data and model design together to resolve training instability issues and achieve competitive performance with GS-LRM. We believe that such experiences are major contributions in the large scale training era. For this reason, we also share all the details about our experiences in the paper, which is valuable to the community, as recognized by reviewer **NfXg**.
> 3. **Training instability**: There are two reasons behind LRM-Zero’s training instability:
>    1. In order to make our synthesized Zeroverse dataset generalize to real 3D data, it needs to be a superset of real-world objects. To achieve this, we need to make Zeroverse harder than Objaverse. GS-LRM, which does not have structured outputs like NeRF-based LRM, is itself prone to training instability issues. The benefits of making Zeroverse harder than Objaverse might not be obvious for the single object reconstruction task, but can open opportunities for other tasks where data is scarce.
>    2. Having too much boolean difference augmented objects in the training data is the main cause of our training instability issues. As shown in experiment 13 in Tab. 3 in the rebuttal pdf, when reducing the ratio of data with boolean difference augmentation, we don’t have training instability issues with the default training hyperparameters from GS-LRM, including the 0.5 perceptual loss weight.
>
>    With the above reasons, we think that the instability is explainable within the current presented framework.
>
> 4. **Qualitative comparison**: In Fig. 8 in our supplementary material, we have included 6 qualitative comparison results from LRM-Zero and GS-LRM on GSO and ABO. As suggested by the reviewer, we have included additional qualitative comparison results on our anonymous website (please check https://lrmzero2024.github.io/page_lrm_zero_vs_gs_lrm.html). The 4th to last row is where LRM-Zero outperforms GS-LRM on objects with detailed texture, since Zeroverse objects use a high-quality texture dataset. The last three rows are from Fig. 8 in our supplementary material showing where LRM-Zero performs worse than GS-LRM when there is invisible region in the input views (3rd to last row) and where they performs similarly when the input views have good coverage (last two rows). The remaining results are challenging samples that LRM-Zero and GS-LRM perform similarly or GS-LRM performs slightly better.
> 5. **Training on both Objaverse and Zeroverse**: As suggested by the reviewer, we conduct experiment 3 in Tab. 1 in the rebuttal pdf. It shows that training on both Objaverse and Zeroverse performs better than Zeroverse only, but worse than Objaverse only. This is likely due to the reasons discussed in 1\. above: the size of Objaverse is adequate for the single object reconstruction task, and the advantages of Zeroverse are not exploited in this task but has high potential in other scenarios.
>
> References
> \[1\] Zexiang Xu, Kalyan Sunkavalli, Sunil Hadap, and Ravi Ramamoorthi. Deep image-based relighting from optimal sparse samples. ACM Trans. Graph., 37(4), 2018\.

---

> > ### Comment · Reviewer_CePT · 2024-08-09
> >
> > Thanks for the authors' reply. After reading the responses, I decided to keep my rating as it would be insightful work for the community.

---

> > > ### Author Response · Authors · 2024-08-12
> > > **Webpage is removed**
> > >
> > > Dear Reviewers,
> > > Per the request of the Area Chairs, we have taken town the webpage for GS-LRM and LRM-Zero comparisons. We will add these comparisons to our final version.
> > > Thanks,
> > > Submission49 Authors

---

### Official Review · Reviewer_NfXg · 2024-07-22

**Soundness:** 3
**Presentation:** 3
**Contribution:** 3
**Rating:** 6
**Confidence:** 4

**Summary:**

The paper explores the feasibility of using procedurally synthesized datasets of 3D shapes to optimize existing large reconstruction models (LRMs), that procure 3D shapes in the form of NeRFs. The data is constructed by 1) sampling several shapes from a set of parametric primitives (cubes, spheres, tori, etc.) with random positions, scales, and rotations; 2) applying randomly sampled textures (from a preselected set of textures); 3) possible application of additional augmentations (changing the curvature of objects, and allowing concave and thin structures). Numerous experiments on existing Objaverse and proposed Zeroverse datasets show that given the correct degrees of the proposed augmentations, existing LRMs can be trained with the proposed synthetic data to produce comparable although slightly worse results, which suggests that existing scarcity of semantically rich 3D data might not be the main limiting factor for the development of reconstruction models since this data is not necessarily needed.

**Strengths:**

* Since the experiments are very computationally demanding, the presented results are valuable (reproducing them is computationally prohibitive for most academic labs).
* A lot of ablation experiments explore the effects of the proposed augmentations suggesting the existence of an optimal amount of 3D shape augmentation for proper training of LRMs.
* The paper is well-written and easy to follow.

**Weaknesses:**

* In terms of the metrics, models trained with the proposed synthetic data are worse but the authors claim that qualitatively it is hard to see the difference. At the same time, there are almost no qualitative comparisons of reconstructed objects from models trained on real/synthetic data in the main text, supplementary, or on the provided anonymous webpage. It would be better to show these and let the readers conclude if they are close.
* One of the main ideas of the paper that the LRMs do not require semantically rich realistic data for training is somewhat successfully challenged in the same paper, given that the LRMs trained for a longer time improve even further, increasing the gap between the proposed synthetic and conventional data.
* Another issue, mentioned by the authors is that the amount of the used augmentations for optimal performance may depend on the test dataset, so the domain gap between the synthetic data and considered test sets may differ, which limits, to some extent, the generalization ability of the proposed models.

**Questions:**

Have the authors tried to perform any similar experiments for significantly different data synthesis pipelines? For example, the space of primitives can be extended with existing non-primitive shapes or with shapes generated by existing shape generation methods. If the semantics do not matter, the primitive set should not be crucially important, but at the same time non-primitive shapes used as base shapes will contain some relevant data patterns needed to capture high-frequency details in other non-primitive shapes from the test sets. Maybe this way, the authors would not need to find a set of perfect parameters for the augmentations to make the method work.

See limitations.

**Limitations:**

While I do not think any ethics review is necessary for this paper, I think it may make sense to disclose the full computational budget for this project (for example in GPU hours spent) to warn any potential readers of the costs of such works.

---

> ### Author Rebuttal · Authors · 2024-08-07
>
> We thank the reviewer for appreciating the value of our work to the community, the comprehensiveness of our ablation experiments, and our writing clarity. We reply to the questions from the reviewer as follows:
>
> 1. **Qualitative comparison results**: In Fig. 8 in our supplementary material, we have included 6 qualitative comparison results from LRM-Zero and GS-LRM on GSO and ABO. As suggested by the reviewer, we have included additional qualitative comparison results on our anonymous wepbage (please check https://lrmzero2024.github.io/page_lrm_zero_vs_gs_lrm.html). The 4th to last row is where LRM-Zero outperforms GS-LRM on objects with detailed texture, since Zeroverse objects use a high-quality texture dataset. The last three rows are from Fig. 8 in our supplementary material showing where LRM-Zero performs worse than GS-LRM when there is invisible region in the input views (3rd to last row) and where they performs similarly when the input views have good coverage (last two rows). The remaining results are challenging samples that LRM-Zero and GS-LRM perform similarly or GS-LRM performs slightly better.
> 2. **LRM-Zero vs GS-LRM at longer training steps**: it is true that the performance gap between LRM-Zero and GS-LRM is enlarged as they are trained for a longer time, as shown in table 8 and 9\. As discussed in Appendix Sec. E, we believe that many of our early exploration on scaling experiments, including the larger performance gap at 2x training steps, suggest that the model converges slower on Zeroverse than on Objaverse. However, we believe that LRM-Zero's competitive performance is still impressive and that this does not undermine the potential of extending this work to other 3D tasks where data is more scarce (see Sec. B Beyond object-level reconstructions).
> 3. **Generalization of our augmentation configuration**: we did not optimize the Zeroverse augmentation configuration for the testing set metric numbers. Instead, Zeroverse aims to reduce the structural gap by introducing augmentations that allows for better coverage of common shape of real-world objects, e.g. boolean augmentation for concave shapes and wireframe augmentation for thin structure shapes, as detailed in Sec. 3.2. As shown in Table 4/5, some augmentations do not improve numerical metrics (e.g., PSNR) but we found that they are essential for human’s visual alignment. Given these philosophies behind, we think that our method should be generalizable and not just overfit the testing set (also evident in Table 9/Fig 5  with diverse testing data; and Table 5 with different model architectures).
> 4. **Extending the primitive set**: Thanks for the suggestion. This is an interesting idea that would ideally reduce the training cost on carefully ablating the shape augmentation design. However, since this requires lots of workload on both data synthesis and model training, we could not test this idea during the rebuttal period. This is a great idea for future work that we will consider.
> 5. **Full computation budget**: Experiments reported in Tab. 1, 2, 3, and 4 take 96 A100 days (we will correct Sec 4). The scaling experiments in Tab. 8 range from 96 to 384 A100 days. The total exploration, including 33 default-scale experiments and 31 scaling-up experiments, is approximately 9120 A100 days. The cost of Zeroverse data generation is approximately 3200 CPU node days. Note that the final model does not cost much, but the data exploration did. We recognized these hidden budgets. Thus, we decided to reveal all technical details regarding stability and release the data creation code. We hope that this can largely reduce the cost in reproducing or following our works.

---

> > ### Author Response · Authors · 2024-08-12
> > **Webpage is removed**
> >
> > Dear Reviewers,
> > Per the request of the Area Chairs, we have taken town the webpage for GS-LRM and LRM-Zero comparisons. We will add these comparisons to our final version.
> > Thanks,
> > Submission49 Authors

---

### Author Rebuttal · Authors · 2024-08-07

We thank the reviewers for their insightful feedback and their recognition of our clear writing, comprehensive experiments, and the novelty and value of our proposed method (i.e. using synthesized data to train a LRM). We will address the missing citations mentioned by **3hvP** in the revision. Please see our responses to the questions from each reviewer below.

---

### Decision · Program_Chairs · 2024-09-25

**Decision:**

Accept (poster)

**Comment:**

Reviewers have unanimously praised the convenience and the performance of the main idea of training an LRM with purely synthetic data. Additional positives include great presentation and the comprehensiveness of the evaluation. On the other hand, the main concerns include worse performance when training only on a dataset of artist-created 3D assets, and the fact that the performance only gets worse when training on Zeroverse+Objaverse or only on Zeroverse. Additionally, reviewers were concerned about a lack of technical contribution.

The AC decides to accept the paper mainly because of the surprising finding that training LRM on purely synthetic data can lead to a very competetive performance. Here, the AC also agrees with authors' response to R-NfXg that such finding can be exploited for obtaining supervision for problems with scarce hand-made ground-truth (e.g. relighting).

Authors are strongly encouraged to include the results of their additional rebuttal experiments in the camera-ready version.